# Real-Time Stream Processing in Social Networks with RAM³S

**Ilaria Bartolini** * and **Marco Patella**

DISI, University of Bologna, 40100 Bologna, Italy; marco.patella@unibo.it

* Correspondence: ilaria.bartolini@unibo.it

**Abstract:** The avalanche of (both user- and device-generated) multimedia data published in online social networks poses serious challenges to researchers seeking to analyze such data for many different tasks, like recommendation, event recognition, and so on. For some such tasks, the classical "batch" approach of big data analysis is not suitable, due to constraints of real-time or near-real-time processing. This led to the rise of stream processing big data platforms, like Storm and Flink, that are able to process data with a very low latency. However, this complicates the task of data analysis since any implementation has to deal with the technicalities of such platforms, like distributed processing, synchronization, node faults, etc. In this paper, we show how the RAM³S framework could be profitably used to easily implement a variety of applications (such as clothing recommendations, job suggestions, and alert generation for dangerous events), being independent of the particular stream processing big data platforms used. Indeed, by using RAM³S, researchers can concentrate on the development of their data analysis application, completely ignoring the details of the underlying platform.

**Keywords:** stream processing; social networks; big data

## 1. Introduction

Analysis of data coming from online social networks is a hot topic that continues to receive the attention of researchers [1,2]. The uses are countless, from the security of citizens [3], to sentiment analysis [4,5] and narrative inquiry [6], to cite a few. Indeed, although the enormous amount of information generated by social networks is a formidable asset for turning data into value, it is rarely utilized due to the lack of suitable technologies for data management [3]. Big data platforms offer opportunities to handle and analyze large amounts of data; however, the facilities they offer are often too raw, as they focus on issues of fault tolerance, enhanced parallelism, etc. To use such architecture effectively for advanced applications, an additional software layer (middleware) is therefore needed to hide the complexity of the underlying platform [7].

In this paper, we focus on the important case of streaming data, i.e., when data flow into the social network continuously and the ultimate goal is to discover novel and unknown information from data sources as they occur, resulting in very limited latency. The problem is also more challenging when multimedia (MM) data are considered, due to their very nature of complexity, heterogeneity, and large size. This makes processing MM streams computationally expensive, so that computing power becomes a bottleneck when a single centralized system has to be used.

In order to enable efficient analysis of large MM streams, we propose scaling out the underlying system by using big data management platforms, as opposed to scaling up the system by exploiting faster, larger, better hardware [8]. The use of already established big data analysis techniques has the evident benefit of shifting the focus away from low-level issues of fault tolerance, replication, storage/transfer of data, etc., so as to make techniques for stream analysis immediately applicable in a distributed environment.

The (RAM$^3$S) framework [9] enables the implementation of any data stream analysis technique on top of existing platforms for big data stream management. (RAM$^3$S stands for real-time analysis of massive multimedia streams.) Researchers seeking to integrate their technique with the underlying big data architecture can thus avoid incurring the overhead of writing an additional ad-hoc interface software layer. To the best of our knowledge, RAM$^3$S represents the first attempt to address the analysis of massive multimedia streams from a general perspective, by abstracting from the details and the complexity of distributed computing.

In [9], we introduced the technical details of the RAM$^3$S framework and exploited its versatility to provide an experimental comparison of three existing big data platforms for stream data processing, namely Spark Streaming, Storm, and Flink. In this paper, we show the generality of RAM$^3$S, proving how it can be easily extended to implement different applications of data stream analysis. In particular, we show the application of RAM$^3$S to four relevant use cases in the context of social networks. We would, however, highlight the fact that the use of RAM$^3$S is possible for any data stream analysis application, including those in the domains of smart cities, automated industry, cyber-security, smart mobility, public health, and so on.

*Outline*

The organization of the paper is as follows:

- Section 2 provides the motivation for our work, presenting examples of modern social network services requiring real-time processing of massive multimedia data streams.
- In Section 3 we introduce the RAM$^3$S framework, detailing its programming interface that allows generalizing it to the specific application at hand.
- Section 4 shows the three different use cases that we implemented on top of RAM$^3$S, proving its wide range of applicability.
- Finally, Section 5 concludes, pointing out interesting directions for further research.

## 2. Motivation

In this section, we provide a number of examples of use of social network services demanding processing of multimedia stream data with low latency requirements.

**Example 1** (Job Suggestion). *ConnectedTo is a service for business and employment that operates through a website and a mobile app. Its main use is professional networking, allowing employers to publish job proposals and job seekers to post their CVs.*

*Alice recently joined ConnectedTo to find a new job: She posts her CV in order to find a selections of job proposals fitting her working experience, training, and skills. The system should match Alice's CV with the available jobs to present her with only the ones she could be interested in: The filtering (and possibly sorting) of job proposals should be performed in near-real-time, in order to allow Alice to review the requested profile and, for example, discover which of her professional contacts can introduce her to the hiring company.*

*On the other side, Bob is an HR manager at JCN, an IT company based in NY, and a long-term user of ConnectedTo: He wants to publish the proposal for a new job at the JCN R&D department. The system should efficiently filter out candidates who do not fit the profile requested by the company and point out the best candidates for the position with very low latency.*

**Example 2** (Dangerous Events Detection). *The Christchurch Call to Action (https://www.christchurchcall.com/) is a political agreement among 48 countries, initiated by New Zealand Prime Minister Jacinda Ardern, fostered by the Christchurch mosque shootings of 15 March 2019. World leaders and technology companies pledged to "eliminate terrorist and violent extremist content online".*

*Being one of the Internet companies that signed the original pledge, the MePipe video sharing service includes, in its Terms of Service Agreement, the following statement:*

> *"Our service includes automated systems to detect and remove abusive and dangerous activity that could hurt the community at large."*

*This clearly requires analysis every single video posted by any MePipe user to discover its appropriateness. For this, Charlie (an IT engineer at MePipe with specialization in machine learning) has developed a deep-learning-based software that can classify any video in real time on a single machine. However, every minute there are over 500 h of video uploaded to MePipe, which makes the centralized approach followed by Charlie clearly impracticable.*

**Example 3** (Clothing Recommendation). *The app 19-Fasteners allows users to share images of clothes, shoes, garments, etc., and to receive buying suggestions related to the posted images. Darla, a young girl who recently subscribed to 19-Fasteners, is strolling through the streets of Milan, when she spots a wonderful pair of shoes in a shop in the Via Montenapoleone fashion district; however, the price largely exceeds her budget.*

*Luckily, the 19-Fasteners service allows users to upload a photo of garments and to receive offers of similar items from clothing sellers. Darla immediately takes a picture of the shoes and posts the image on the app, specifying her requested price range; in only a few moments, 19-Fasteners returns to Darla a list of items satisfying her budget, in decreasing order of similarity with respect to the uploaded photo.*

*Each server is able to process Darla's request in 500 milliseconds, however managers at 19-Fasteners foresee that, in the near future, the site activity will be about 1K requests/second. Emily, responsible for IT services at 19-Fasteners is now requested to estimate the number of new computers that should be installed at the 19-Fasteners server factory.*

In all the above examples, the input data can be represented as a stream of MM objects: Text (CVs/job proposals), videos (the MePipe casts), and images (clothing pictures). Each input object has to be matched against a knowledge base composed of objects of the same type to produce a classification (inappropriate video) or a ranked list of DB objects (relevant jobs/CVs/clothes). Processing a single object might be a simple task, but the bottleneck is represented by the rate of arrival of the input data, which makes processing of the stream impracticable on a single server. Streaming big data platforms, like Apache Storm and Apache Flink, are indeed able to efficiently parallelize the data processing, dealing with issues peculiar to distributed computing like synchronization, scheduling, data distribution, fault tolerance, load balancing, and so on. The downside of adopting big data platforms is that it is not always straightforward to scale out on top of them a data analysis technique conceived for a centralized scenario. Indeed, the complexity of such architectures makes it hard, for users not expert in distributed systems, to interface their top data stream analysis applications to the bottom data processing layer. To overcome this gap between stream applications and distributed systems, RAM³S was proposed [9] with the goal of facilitating the implementation of techniques for the analysis of multimedia streams on top of big data platforms, by providing a "middleware" software layer, whose top interface is both simple (hiding peculiarities of the bottom data layer) and general (allowing any data stream application to be effectively scaled out). In this way, the user can concentrate on details of stream analysis, avoiding the overhead of understanding issues peculiar to distributed systems.

## 3. Background

This section contains the necessary background information on RAM³S, including a description of its top-level interface that, when appropriately instantiated, can be used to implement the different applications that will be described in the paper.

The RAM³S framework offers a general infrastructure for multimedia streams analysis on top of big data platforms. Through the use of RAM³S, researchers can effectively scaling out methods originally conceived for a centralized scenario, by applying their multimedia analysis algorithms to very large data streams, without paying the cost of mastering technicalities typical of distributed

systems. To be both effective and general, we designed RAM$^3$S as a "middleware" software layer between the top data stream application and the underlying big data platforms.

The kernel of RAM$^3$S is composed by two interfaces that, when appropriately instantiated, can act as a general tool for data streams analysis: The `Analyzer` and the `Receiver` (see Figure 1, where the result of the analysis is depicted as an alarm, but results of different types are also supported).

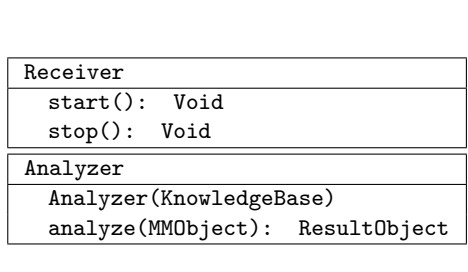

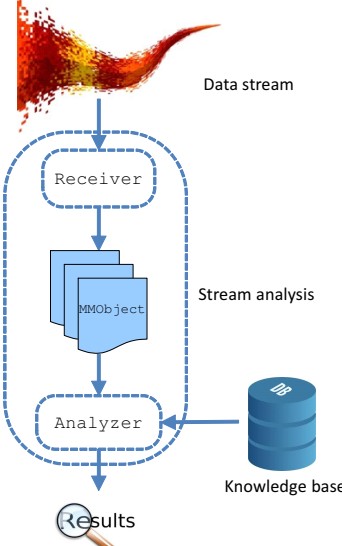

**Figure 1.** General view of the two main RAM$^3$S classes (adapted from [9]).

The `Analyzer` component is responsible for analyzing individual multimedia objects (instances of the `MMObject` class). In particular, data in the underlying knowledge base are compared with all incoming multimedia objects to compute the result of its analysis. On the other hand, the `Receiver` should decompose a single input multimedia stream into a sequence of individual `MMObjects` that will be fed one by one to the `Analyzer`. The interface of the `Receiver` component is quite simple, as only two methods should be defined: The `start()` and the `stop()` methods, respectively, specify how to initiate and terminate the acquisition of the data stream. As soon as it is created, the `Receiver` component outputs a sequence of instances of `MMObject`. The `Analyzer` component only contains the constructor and the `analyze()` method. The constructor has a single parameter, representing the knowledge base which will be used for analyzing individual objects. The `analyze()` method, on the other hand, takes a single instance of `MMObject` in input and outputs a `ResultObject`, computed as the result of "matching" the input `MMObject` with the knowledge base.

The basic RAM$^3$S assumption is that the input stream of multimedia data can be implemented by way of a sequence of individual `MMObject` instances: The result is obtained by way of the continuous analysis of such `MMObjects`, repeated over time. We believe that this reductionist approach does not turn out to be simplistic, as its objective is not to represent all possible MM stream analysis techniques. Indeed, the goals are generality, to embody a sufficiently large majority of algorithms for stream analysis, and simplicity, to make it as easy as possible to interface with the underlying big data platform.

*3.1. Interface to Big Data Platforms*

The RAM$^3$S software framework (and its application to specific use cases) can be immediately applied in a centralized environment, by creating a `Receiver` instance for each input data stream and connecting them to a single `Analyzer` instance in charge of collecting outputs of the `Receivers`. In order to effectively move towards a distributed scenario, in the following we detail how the framework is interfaced to the big data platforms considered by RAM$^3$S, namely Apache Spark Streaming, Apache Storm, and Apache Flink. These are depicted in Figure 2, where we emphasize the possibility of

receiving the input data from multiple concurrent streams, while the final output is collected in a single "result" node.

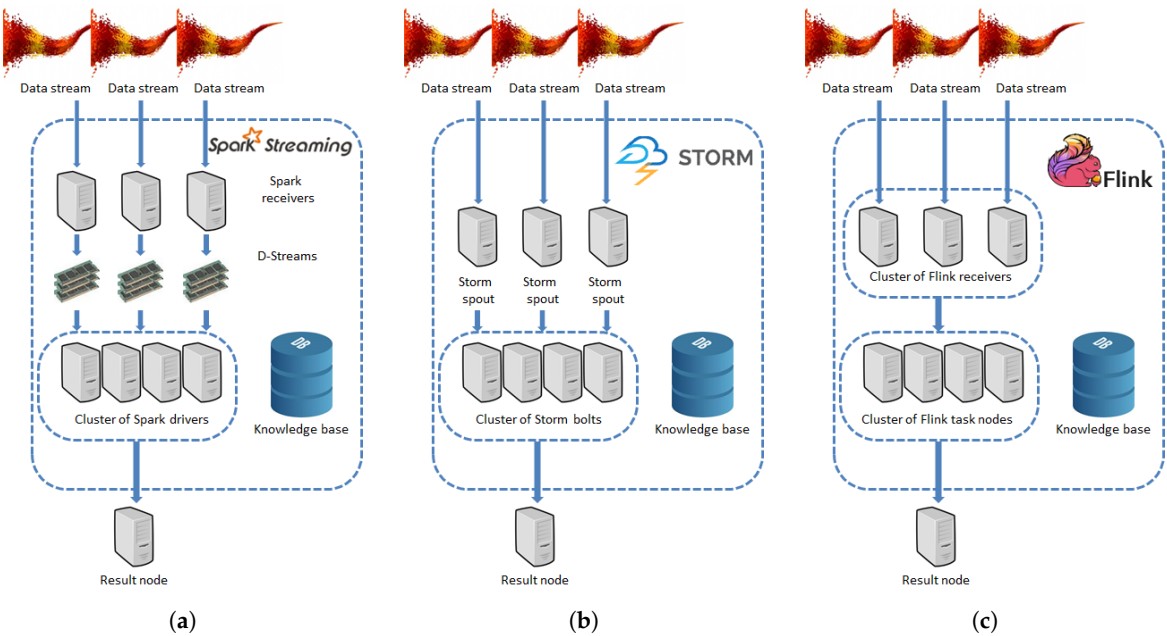

**Figure 2.** Architecture for big data platforms: Spark (**a**); Storm (**b**); and Flink (**c**).

### 3.1.1. Apache Spark

Apache Spark (https://spark.apache.org) [10] is an open source platform, originally developed at the University of California, Berkeley's AMPLab. The main idea of Spark is to store data in a resilient distributed dataset (RDD), representing a fault-tolerant read-only collection of (Python, Java, or Scala) objects that can be stored in main memory and are partitioned across several machines. RDDs are immutable and their operations are lazy. The sequence of operations that produced a particular RDD (its "lineage") is stored within the RDD itself; in this way, the RDD can be reconstructed in the event of failures, thus ensuring fault-tolerance. Spark applications are thus just sequences of operations on RDDs, having the final objective of producing the desired result. Two types of operations can be performed on an RDD: A new RDD can be built through a transformation operation, producing it from an original input RDD; on the other hand, any operation producing a single value or writing an output on disk is called an action. The kernel of Spark is Spark Core, offering tools for the dispatching and scheduling of distributed tasks. Any program would use a functional programming model to invoke operations on Spark Core. As a consequence, Spark Core schedules the execution of the function in parallel on the machine cluster. The overall execution assumes the form of a directed acyclic graph (DAG), with operations on RDDs performed on a computing node representing nodes and dependencies among RDDs representing arcs. The Streaming version of Spark introduces the receiver as a new component, responsible for receiving input data from a variety of sources, like TCP/IP sockets, Apache Kafka, or Twitter. Real-time processing is now available through the introduction of a new element, the D-Stream, consisting in a sequence of RDDs sampled every *n* time units. The Spark architecture is able to accept any D-Stream as a "regular" RDD to be processed: This mode of operation is called micro-batching. Any Spark Streaming application consists of two main components: Receivers and drivers. The data acquisition logic is specified through a receiver, by defining the `OnStart()` and `OnStop()` methods to initiate and terminate the data input, respectively; each receiver resides in a core in the architecture. On the other hand, the program specifies in a driver the receiver(s) producing the D-Streams and the sequence of transformations over the data, terminating with an action to store the final result.

In RAM$^3$S (Figure 2a), every input stream is connected to a `Receiver`, which corresponds to a single Spark receiver: The receiver output is then buffered so as to create the D-stream. The buffer is composed by $n$ `MMObject` instances, $n$ being a system parameter. A D-stream is built and sent to the cluster of Spark drivers every time the buffer is full. The `start()` and `stop()` methods of the `Receiver` interface correspond to the `OnStart()` and `OnStop()` methods of the Spark receiver. An `Analyzer` is included in every Spark driver. The logic of Spark drivers will iterate the `analyze()` method over all the $n$ `MMObject` instances contained in the input D-stream, creating a sequence of $n$ `ResultObjects`. Finally, the result sequence is sent to the result server.

### 3.1.2. Apache Storm

Apache Storm (https://storm.apache.org) is an open source platform that was developed originally by Backtype and later sold to Twitter. The subject of computation in Storm is the tuple element, a serializable user-defined type. The Storm architecture consists of nodes of two different categories: A Spout is a node producing a tuple stream, connecting to an external data source (such as a message queue or a TCP/IP socket), while a node performing stream analysis is called a bolt. Ideally, each bolt only performs a transformation of limited complexity; in this way, several bolt nodes cooperate in a coordinate fashion to perform the entire computation. Every bolt contains two methods: The `prepare()` method is used when the bolt is initiated, while the `execute(Tuple)` is invoked whenever a `Tuple` object is available for processing. A Storm application can be defined through a topology of spout and bolt nodes forming a DAG, with arcs representing streams of tuples flowing from one node to another. Therefore, the computation pipeline is defined by the overall opology. One of the Storm fundamental concepts is acknowledgment of input data, which is the key tool for providing fault-tolerance; in this way, if a node emits a tuple which is not acknowledged, it can be re-processed, for example by forwarding the tuple to another node. This corresponds to the "at-least-once" semantics, ensuring that all data are always processed. This allows for a limited latency, but does not ensure that the the final result is unique (since some data might have been processed more than once).

Just as with Spark, each stream is sent to a RAM$^3$S `Receiver`, which is embedded in a Storm spout (Figure 2b). However, in this case each `MMObject` produced by the spout is sent to the cluster of Storm bolts immediately. Every Storm bolt corresponds to a single `Analyzer` instance. The `prepare()` method is matched with the `Analyzer` constructor, while `execute()` method is equivalent to `analyze()`, that has to be executed for all received `MMObjects`. Any time a `MMObject` is received by the `Analyzer`, the latter should acknowledge the emitting `Receiver` in order to respect the semantics of Storm. Compared to the Spark solution, we now have the benefit of a real stream processing architecture: It is likely that latency is reduced, as we do not have to wait until a D-stream is completed to start analyzing any single `MMObject`.

### 3.1.3. Apache Flink

Apache Flink (https://flink.apache.org) is an open source platform, which begun as a German research project [11] and was subsequently transformed into an Apache top-level project. The core of Flink is a distributed streaming dataflow, accepting programs structured as graphs (JobGraph) of activities that produce and consume data. Each JobGraph can be executed using one of the different distribution options available for Flink (like single JVM, YARN, or cloud). By switching data buffering on and off, Flink is able to perform both batch and stream processing. The topology state is saved by Flink at fixed time intervals either in main memory or on Distributed File System. Flink processes data according to the snapshot algorithm [12], which uses marker messages to save the state of the entire distributed system, avoiding loss and replication of data. The use of the snapshot algorithm allows Flink to ensure that data are processed at least one time and no more than that, thus obtaining the "exactly-once" semantics. Clearly, this helps in reducing both the latency and the overhead, also maintaining the original data flow. Finally, Flink has a "natural" data flow control, because of its use of fixed-length data queues to transfer data between nodes of the topology. The main difference between

Storm and Flink is that, in the former, the programmer is responsible for defining the graph topology, while the latter does not have this requirement, as each node of the JobGraph can be deployed on any single computing node in the architecture.

The RAM$^3$S (Figure 2c) configuration for Flink appears to be rather similar to the Storm case: Streams are sent to a cluster of Flink receivers, each encompassing a single `Receiver` instance that sends individual `MMObjects` to the cluster of Flink task nodes. Any Flink task node contains an `Analyzer`, which executes the `analyze()` method for any received `MMObject` and computes its final result.

With respect to the Storm architecture, although in Figure 2c Flink receivers are depicted separately from Flink task nodes for ease of presentation, we would like to point out that this distinction is only virtual, since each computing node can execute any activity of the Flink JobGraph, i.e., no node is explicitly designed as "receiver" or "task"; on the contrary, any single computing node can host, at any given moment, either a receiver or a task node. In this way, the Flink architecture is able to achieve high resiliency and versatility, because (i) node faults can be easily accommodated for both receivers and task nodes, and (ii) input data streams can be added/deleted on demand (while in Storm this would require modifying the topology, which necessitates restarting the computation).

## 4. Use Cases

In this section we describe the four different applications that have been implemented on top of the RAM$^3$S framework, providing details on the effort required to instantiate the RAM$^3$S classes.

The UML diagram of the main classes of the RAM$^3$S framework is depicted in Figure 3. As said, the `Receiver` and the `Analyzer` interfaces have to be properly implemented by the `ApplicationReceiver` and `ApplicationAnalyzer` classes. The same happens with the `iFactory` interface that is implemented by the `ApplicationFactory` class that instantiates the `ApplicationReceiver` and `ApplicationAnalyzer` classes. The last abstract class that needs to be properly extended is the `MainFramework` class: For this, the programmer could exploit the `SparkFramework`, `StormFramework`, and `FlinkFramework` classes (not shown in Figure 3), providing methods for creating nodes and start services specific to each framework. Such classes are the key to hide details of the bottom big data platforms to the user: All issues specific to distributed data processing are dealt with within these classes, that provide the programmer with a simple interface using the classes depicted in Figure 3. The simplicity of the RAM$^3$S interface demonstrates our thesis: Using RAM$^3$S the intricacies of underlying frameworks (like fault-tolerance, isolation, and stateful processing) are hidden to the user, who can focus on how to adapt her data stream analysis technique to the `Receiver` and the `Analyzer` interfaces.

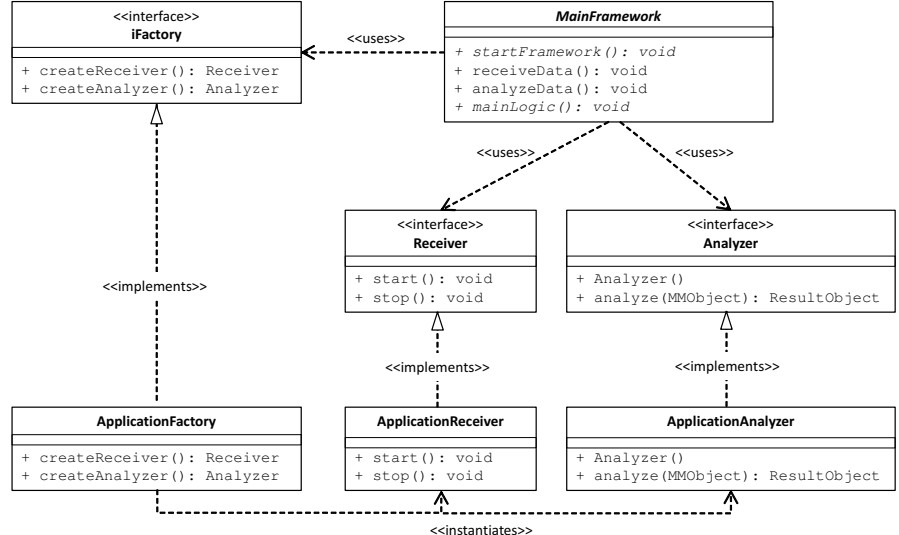

**Figure 3.** UML diagram for the main classes of RAM$^3$S.

### 4.1. Experimental Setup

All experiments described in the following sections have been performed in our datalab, (http://www-db.disi.unibo.it/research/datalab) equipped with a cluster of 16 PCs with an Intel Core 2 Duo, 2.80 GHz CPU and 4 GB of RAM hosting Spark/Flink workers and Storm bolts, connected with a 100 Mbit/s Ethernet network. An additional identical machine was designated as the master, containing the Spark receiver, the Storm spout, and the Flink job tracker. Finally, an external machine hosts the RabbitMQ (https://www.rabbitmq.com/) message broker, which is used to simulate the input stream. The experimental environment is illustrated in Figure 4.

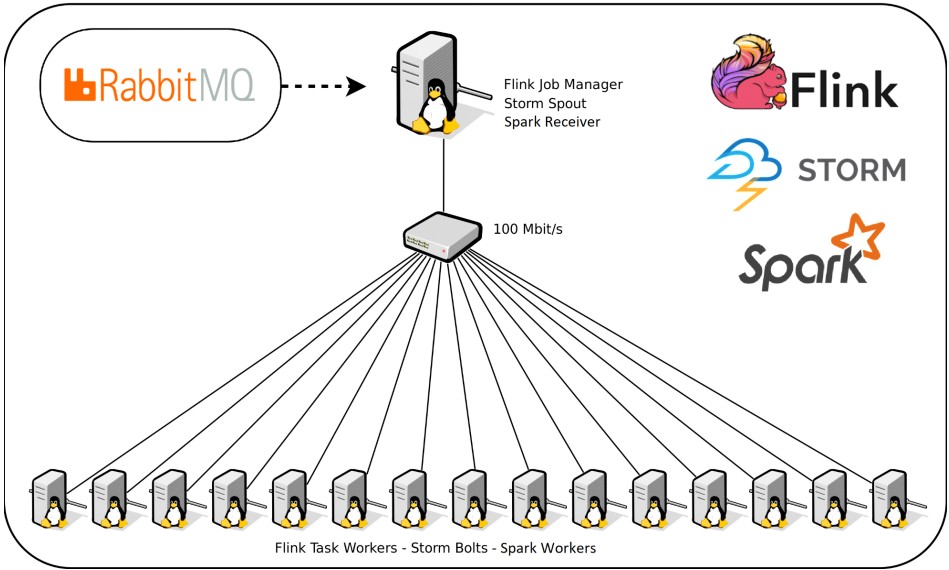

**Figure 4.** The experimental environment of datalab, where all use cases were implemented.

### 4.2. Face Recognition

Automatic face recognition was the use case first developed in RAM$^3$S [9].

The goal of a face recognition system is to identify a person from a digital image. Usually, a face recognition system works by comparing features automatically extracted from the input image with those previously extracted from faces in the knowledge base. Applications for face recognition range from access control in security systems to human-computer interaction to identification of suspects to search for potential criminals and terrorists: A relevant example include the use of facial recognition technology by the police in Tampa Bay, Florida at Super Bowl XXXV in January 2001, although the effectiveness of such technology was extremely limited [13], while more recent uses are likely much less publicized due to the necessary secrecy involved.

Our use case is a system for detecting the presence of known criminals from video streams coming from cameras scattered on the territory. This requires to perform four main steps in sequence: Frame splitting, face detection, face recognition, and result check.

1. Frame splitting consists of separating into frames the video coming from the camera(s), producing one (or more) sequence of images.
2. During face detection, each image of the sequences is analyzed to check whether it contains a face.
3. In case a face is discovered, the recognition phase compares it against a number of known faces, to retrieve the known face most similar to the discovered face.
4. In case the similarity between the discovered face and its most similar known face is sufficiently high, the face is considered as correctly recognized, otherwise it is regarded as an unknown face. For the purpose of suspect identification, whenever a discovered face is sufficiently similar to one of the faces in the knowledge base, an alarm is raised.

For face detection, we used the well-known Viola-Jones algorithm [14], while for face recognition a technique exploiting principal component analysis using eigenfaces was exploited [15]. For both alternatives we used implementations provided by the OpenIMAJ library(http://openimaj.org/).

Figure 5 shows a few images taken from the RAM$^3$S prototype, where examples of correctly recognized faces and of unknown faces are shown. For each image, the system detects the position of the face and retrieves the most similar face in the knowledge base ("Response Subject Name" in each example). If the similarity between the input image and its closest knowledge base image is high enough, then the face is considered recognized (examples (a) and (b)), otherwise the person is considered unknown (examples (c) and (d)). Each example also reports the actual identity of each image ("Real Subject Name" field).

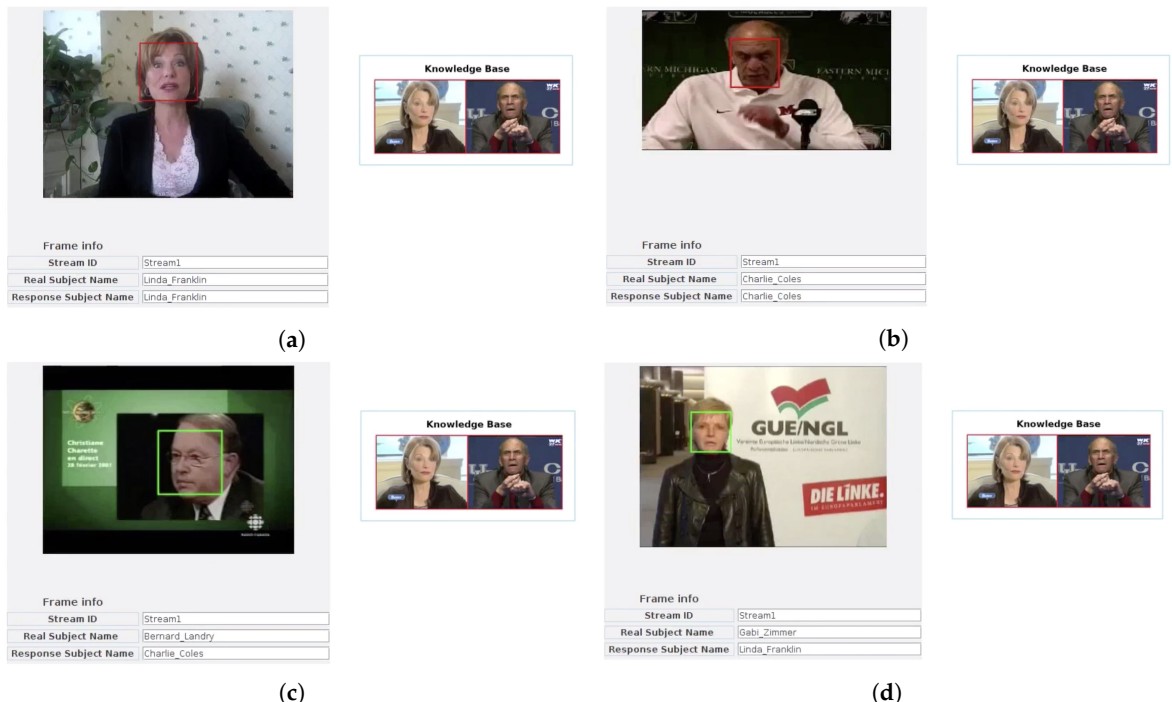

**Figure 5.** Face recognition use case demo: Examples of positive (**a**,**b**) and negative (**c**,**d**) generated alerts with respect to the provided knowledge base.

The effectiveness of RAM$^3$S in reducing the effort required to distribute an application originally conceived for a centralized scenario is confirmed by the number of lines of code required to implement the distributed face recognition system. Originally, every single implementation for Apache Spark, Apache Storm, and Apache Flink required about 170, 150, and 200 lines of code, respectively. By using RAM$^3$S, we built the whole system with only 124 lines of Java code. The benefits of RAM$^3$S are clear:

**Simplicity:** The user does not have to directly interface with the code of the three frameworks.
**Generality:** The user is able to choose at runtime which of the three big data platforms she wants to use.
**Efficiency:** The overall number of lines of code is reduced (124 vs. about 500).

Using this scenario, we performed an additional experiment comparing the relative performance of Apache Storm and Apache Flink. For this experiment and the one included in Section 4.3, we intentionally left out Apache Spark, since its performance is known to be inferior to that of Storm or Flink, due to the the management of D-streams, making Spark not a "real" streaming engine [16].

Our real dataset consists of the YouTube Faces Dataset (YTFD) [17], including 3425 videos of 1595 different people, with videos having different resolutions (480x360 pixels is the most common size) and a total of 621,126 frames containing at least a face (on average, 181.3 frames/video). The average dimension for an image is 22.3 KB.

The input data stream was simulated by reading images from the file system local to each machine and looping through them for a specified amount of time (3 min for the performed experiment); finally, the result was sent to the master node. We chose this modality of input feeding because the amount of data would have rapidly saturated the network (see the experiment described in Section 4.3). The throughput of the overall system can then be computed as the total number of images analyzed by the cluster nodes divided by 180. The experiment was then repeated by switching off a specified number of cluster nodes.

Results are plotted in Figure 6, where we also show a linear regression function, proving the linear scalabitlity of the overall system.

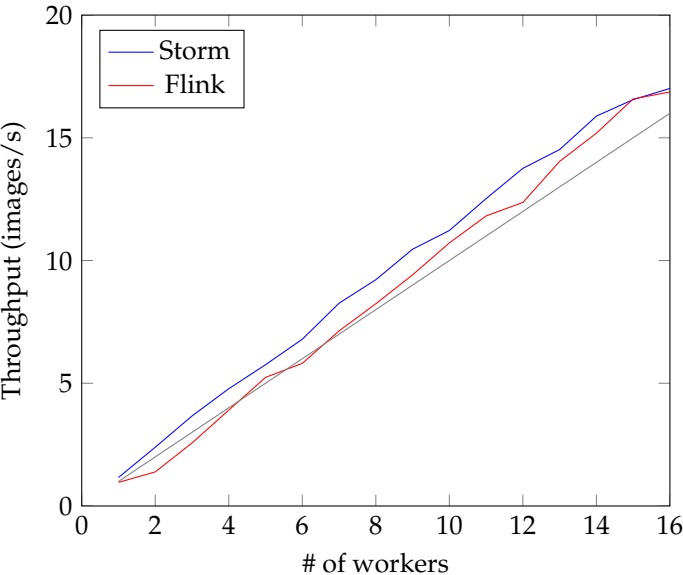

**Figure 6.** Throughput for Storm and Flink.

As to the comparison between Storm and Flink, our results confirm those already presented in [9], although on a different, slightly better performing, experimental environment: Storm attains a better performance than Flink, since the latter pays a superior flexibility and the exactly-once semantics with a slightly diminished efficiency.

### 4.3. Plate Recognition

Automatic license plate recognition (ALPR) concerns the identification of vehicle license plates from an image or video [18]. ALPR was initially developed in the UK during the 1970s (https://en.wikipedia.org/wiki/Automatic_number-plate_recognition). Applications of ALPR range from electronic toll collection on pay-per-use roads to speed limit enforcement and traffic control. The key technology for ALPR is optical character recognition (OCR) on images taken by cameras. The relevance of ALPR is also demonstrated by the fact that many countries recently favored it by adopting retro-reflective plates or using fonts that can be more easily interpreted by OCR.

ALPR algorithms typically follow these steps to perform plate identification:

1. **Plate localization:** To discover the plate in the image.
2. **Plate orientation:** To correct the possible skewing of the plate.
3. **Character segmentation:** To detect actual characters within the plate.
4. **OCR:** To recognize the extracted characters.

ALPR can be performed directly on the camera or on an external server. The former solution clearly requires a higher cost and its accuracy is likely to be inferior to the latter: For this, we consider

that images are taken by a number of cameras scattered throughout the territory and sent as a data stream to an ALPR system, exploiting distributed computing to take care of the amount of input data.

Our implementation used OpenALPR (http://www.openalpr.com/), an open source ALPR library written in C++, exploiting OpenCV (https://opencv.org/) for image processing and Tesseract (https://github.com/tesseract-ocr) for OCR. The input data stream was simulated by feeding car images to a RabbitMQ queue, which was then connected to the implemented `Receiver`. Car images were taken from the OpenALPR benchmark (https://github.com/openalpr/benchmarks/tree/master/endtoend/), which includes pictures of cars taken under different light conditions with resolution with ranges $640 \times 480$, $800 \times, 600$, and $1792 \times 1312$ pixels. The average image size is 368 KB with a few images having size over 1 MB. As to the effort required to implement this use case, the total number of lines of code was around 100 (this was the first use case implemented natively in RAM$^3$S, thus no comparison is possible with the effort required to implement the system on top of the three big data platforms separately).

Figure 7 shows some visual examples taken from the RAM$^3$S application. As for the previous use case, each example shows the actual and the recognized plate ID ("Real Subject Name" and "Response Subject Name" field, respectively) and only plates included in the knowledge base are highlighted in red (examples (a) and (b)).

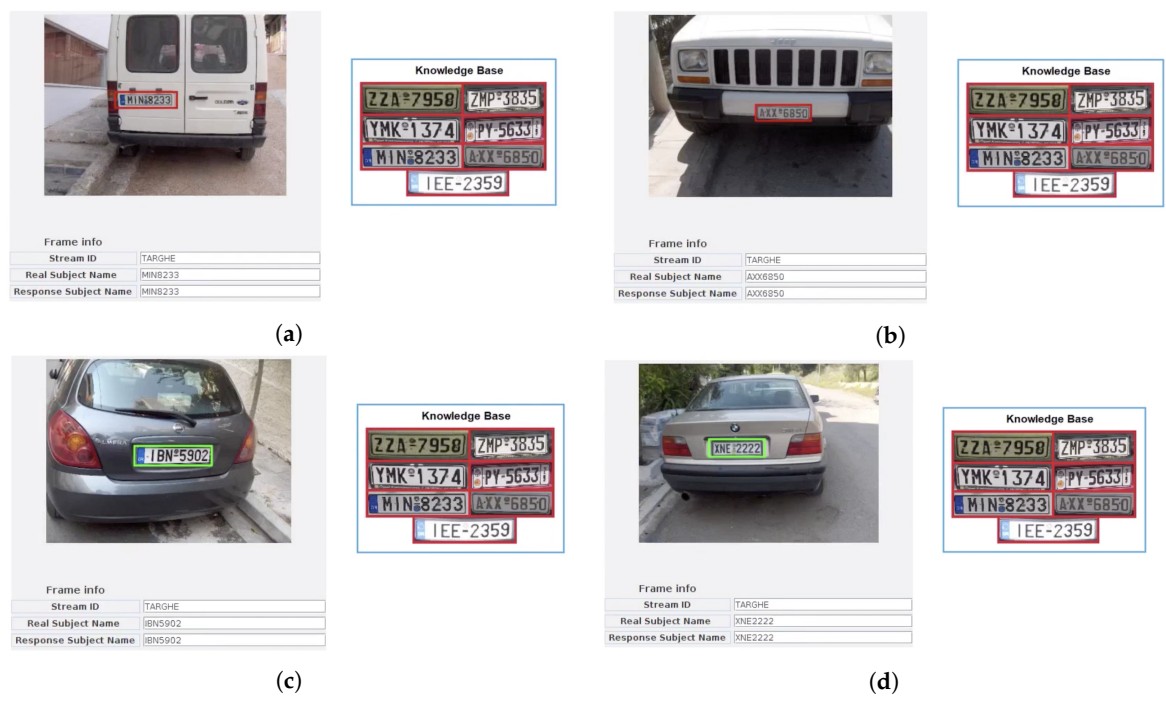

**Figure 7.** Plate recognition use case demo: Examples of positive (**a**,**b**) and negative (**c**,**d**) generated alerts with respect to the provided knowledge base.

This implementation also allowed us to perform an experiment comparing the efficiency of Flink and Storm in scaling out the stream analysis. The experimental setup was the same used in the face recognition use case. We measured the sustainable input rate by increasing the frequency with which images are fed into the RabbitMQ queue and monitoring its length. When the overall system is overloaded, the queue starts filling up and input images overflow: The previously used frequency is taken as the sustainable input rate.

Results for Storm and Flink are included in Figure 8. Graphs confirm results obtained for the face recognition use case: Storm attains a slightly better performance than Flink. We then performed an additional experiment, to find bottlenecks of the system: We purposely saturated the network connecting computing nodes (which has a transfer rate of 100 Mbit/s) by feeding into the queue only

images with size larger than 1 MB. Results on the Storm platform, named "Storm-large-files" in the graph, demonstrate that the system saturates at an input rate of eight images per second, corresponding to a network traffic of more than 64 Mbit/s, which is consistent with the peak transfer rate of the connecting LAN.

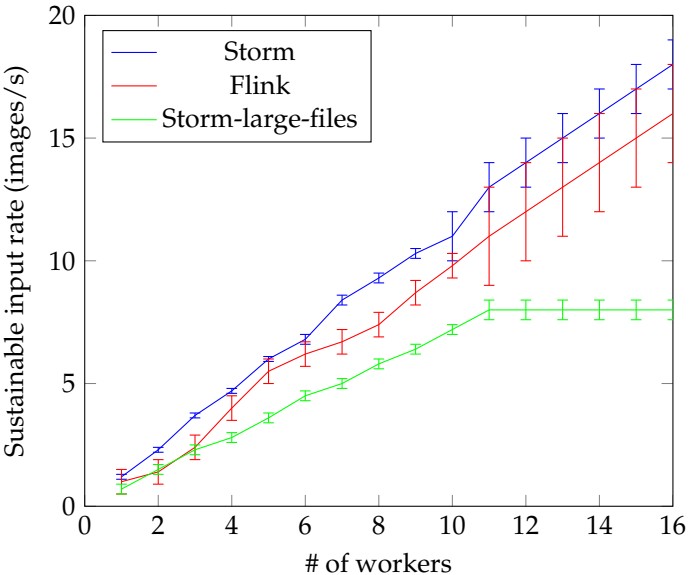

**Figure 8.** Sustainable input rate for Storm and Flink.

*4.4. Printed Text Recognition in Videos*

This use case is presented in the context of Example 2. In particular, we extended RAM³S to automatically extract printed text within video streams. The objective was twofold:

1. Identifying "critical" videos by analyzing and automatically interpreting the streams of a significant data sample.
2. Defining new useful services in the context of monitoring proselytizing phenomena by terrorist groups.

Figure 9 shows the implemented steps for automatically extracting printed text from videos:

1. The first summarization step aims to eliminate all superfluous frames, maintaining only those relevant to the purpose of the final application, thus only key frames in which true information is present are retained. This was obtained with a frame-to-frame analysis, eliminating frames that are too similar to each other. The summarization process exploits the functionalities of the SHIATSU video retrieval tools [19], based on HSV color histograms [20] and the edge change ratio (ECR) [21].
2. When the summary has been obtained, the image analysis phase takes place by considering the selected key frames only. Each image is first filtered to segment text and logos/symbols from the background, then OCR is performed by using the Tesseract library, while logos are extracted using the OpenIMAJ library.
3. Once the text detection phase is over, extracted text and logos are compared with those included in the knowledge base, so as to recognize the ones that have been considered critical.

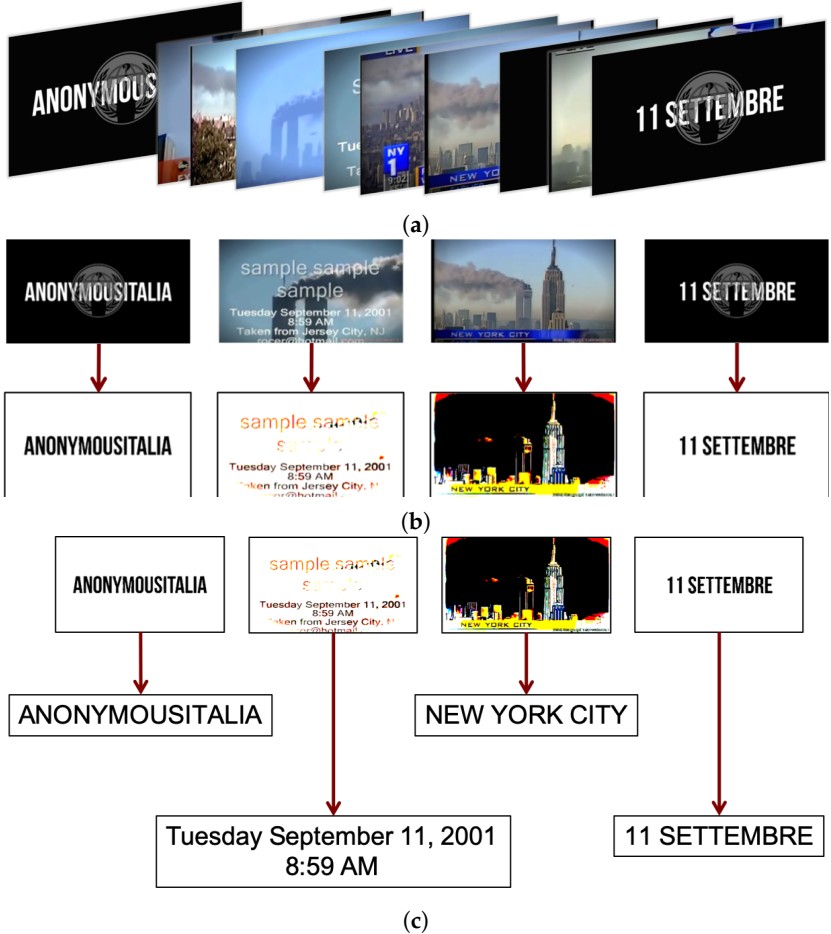

**Figure 9.** Printed text recognition in videos use case: Video summarization (**a**); noise filter application (**b**); text recognition (**c**).

The implementation of this use case in RAM$^3$S, which required about 150 lines of code, allowed us to compare the performance of the three underlying big data platforms in analyzing videos. In particular, in order to provide the system with a stable input stream, the summarization step was performed offline (note that this is the less CPU-demanding step among those required in this use case) and only relevant key frames were fed to the RabbitMQ queue. Key frames were extracted from a dataset of about 100 YouTube videos dealing with terrorism and proselytism. The sustainable input rate (computed as in Section 4.3) for Spark, Storm, and Flink is shown in Figure 10a, while Figure 10b shows the speedup, computed as $S_n = IR_n/IR_1$, where $IR_n$ is the measured sustainable input rate on $n = 1, 2, \ldots$ cluster nodes.

The graphs again confirm results obtained in [9]: Spark performance is the worst among all platforms, while Storm is slightly faster than Flink. Results for Storm and Flink also exhibit a bottleneck when reaching the input rate of 100 images/second: This is again due to the network speed which saturates around 64 Mbit/s (the average size of images in our dataset is 80 KB). As to the speedup achievable when increasing the number of machines, all platforms display a quasi-ideal speedup when the number of nodes is limited (compare with the ideal gray line in Figure 10b). When increasing the cluster size, the system scalability begins to diverge from linearity: The system that first presents such behavior is Storm, due to the network bottleneck.

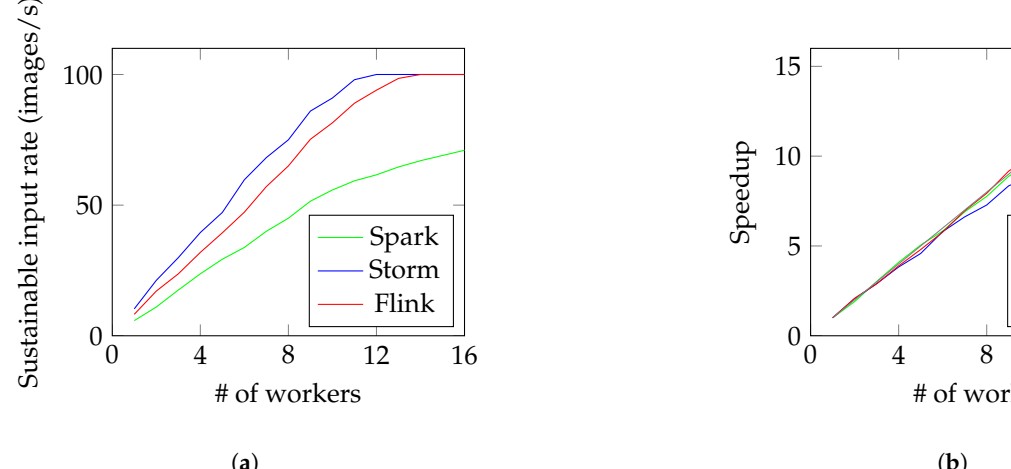

**Figure 10.** Sustainable input rate (**a**) and speedup (**b**) for Spark, Storm, and Flink.

*4.5. Shoes Classification*

The last implementation of RAM³S we present refers to Example 3. In particular, we used the UT Zappos50K dataset (http://vision.cs.utexas.edu/projects/finegrained/utzap50k/) [22] to classify shoe images into one of the main categories in the dataset. Figure 11 shows some examples for the boots, high heels, and sandals classes, displaying the real and predicted category ("Real Subject Name" and "Response Subject Name" field, respectively).

For classification a 1-NN classifier was used, with similarity computed by using features automatically extracted with the WINDSURF library (http://www-db.disi.unibo.it/Windsurf/) [23]. The implementation, which required about 120 lines of code, displays the dataset image closest to the submitted input image (see Figure 11): This mimics the scenario of Example 3, where the system has to provide the user with a recommendation of garments which are mostly similar to the input one.

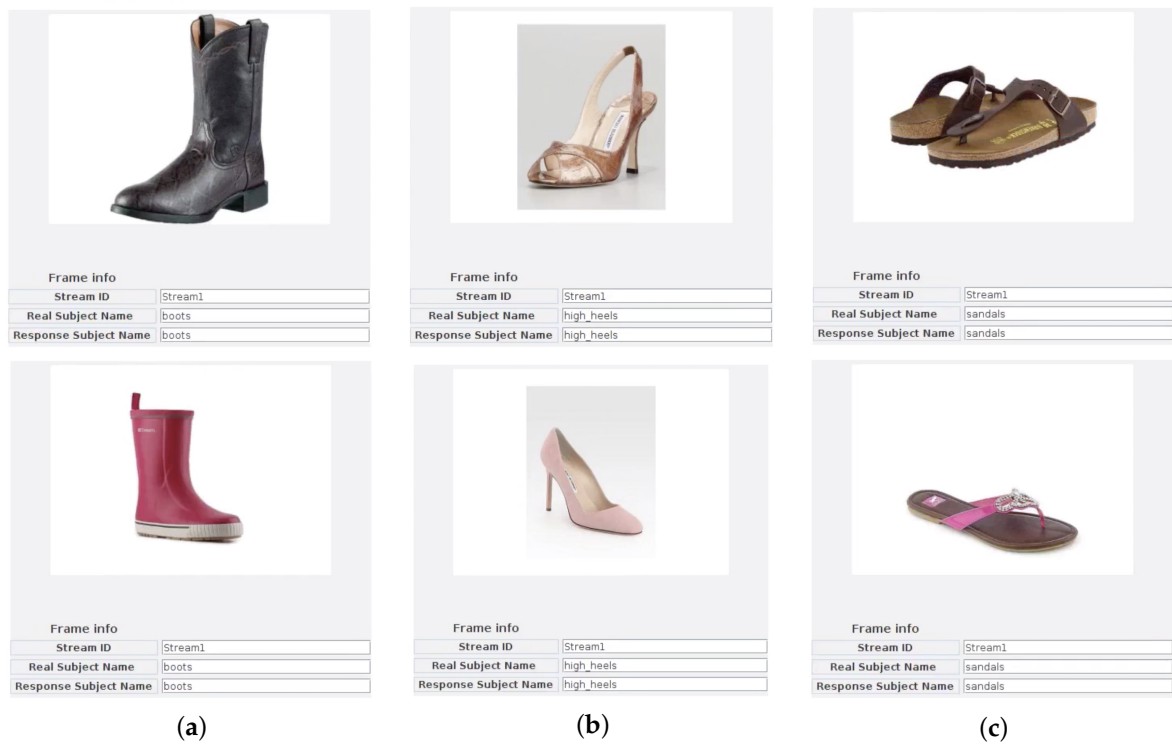

**Figure 11.** Shoes classification use case demo: Classification examples for boots (**a**); high heels (**b**); and sandals (**c**).

Results comparing performance of Spark, Storm, and Flink for this use case exhibit a similar behavior with respect to those obtained in Sections 4.2–4.4; therefore, they are not included here.

## 5. Conclusions

Analysis of data in online social networks usually requires the use of big data platforms, so as to deal with characteristics of volume and velocity of data. In particular, in this paper we have considered the relevant case of streaming data, where incoming data are to be analyzed in real time, in order to consider their freshness, and with requirements of very low latency. However, including platforms for analysis of streaming big data requires considering details of distributed computing, which can be not so obvious for researchers specializing in data analysis. To overcome these difficulties, we have recently introduced the RAM³S framework, which is able to hide details of underlying platforms by exposing to the programmer a very simple interface. We have demonstrated the generality and ease of use of RAM³S by presenting four different use cases that have been implemented by appropriately instantiating RAM³S classes. We believe that the use of the RAM³S framework can effectively help researchers in scaling out to distributed computing scenarios analysis techniques that were initially conceived for a centralized system. Among the future research directions that we are interested in pursuing, we put forward the application of RAM³S in other contexts (like automated industry, smart mobility, and public health) and expanding it to other, recently introduced, big data platforms, like Apache Samza (https://samza.apache.org) [24].

**Author Contributions:** Conceptualization, I.B. and M.P.; Data curation, I.B. and M.P.; Formal analysis, I.B. and M.P.; Investigation, I.B. and M.P.; Methodology, I.B. and M.P.; Supervision, I.B. and M.P.; Validation, I.B. and M.P.;Writing original draft, I.B. and M.P.; Writing review editing, I.B. and M.P.

**Funding:** This research received no external funding.

**Conflicts of Interest:** The authors declare no conflict of interest. The funders had no role in the design of the study; in the collection, analyses, or interpretation of data; in the writing of the manuscript, or in the decision to publish the results.

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
