# Peer review of "Real-Time Stream Processing in Social Networks with RAM3S"

_futureinternet, doi:10.3390/fi11120249_

Round 1

Reviewer 1 Report

The aim of the article is introducing a “middleware” software layer between the underlying Big Data platforms and the top data stream application
in order to hide the complexity of underlying platforms.

The idea of implementing this middleware is very interesting.
Moreover, the article is very well written in a good English and the topics of the article fit well the topics of the journal.

It would be interesting to understand if this solution can be applied, in addition to the social network, in other fields;
for example to manage the data in motion in industrial fields.
This could be an interesting point to be reported in the conclusion, paired with further developments planned for RAM3S.

The major weaknesses of the article is that authors do not clearly explain the motivation behind their research work.
Specifically, it is not clear the statement at row 22: "the services they provide are often too raw"
Furthermore, the motivations scenarios reported in section 2 justify the use of Big data platforms but not the need for a new layer such a middleware on top of the Big data platforms.

In addition, the use cases under section 4 do not demonstrate the real benefits deriving from the implementation of RAM3S (except for reducing the number of lines of code as written on line 287)
against the drawback to create manage the complexity of a new layer.

For the above reasons, I advise the authors to add more details about the complexity of Big Data Platforms that requires to should be managed with higher-level middleware.

Finally, the authors should also specify if there other solutions available in the literature that aim to face the issue of wrapping the underlying Big Data platforms

Author Response

The aim of the article is introducing a “middleware” software layer between the underlying Big Data platforms and the top data stream application in order to hide the complexity of underlying platforms.

The idea of implementing this middleware is very interesting. Moreover, the article is very well written in a good English and the topics of the article fit well the topics of the journal.

It would be interesting to understand if this solution can be applied, in addition to the social network, in other fields; for example to manage the data in motion in industrial fields. This could be an interesting point to be reported in the conclusion, paired with further developments planned for RAM3S.

R: Yes, RAM^3S can be applied to any case where massive multimedia data streams need to be analyzed in real time. We highlighted this fact in the revised version both in the introductory section (lines 48-50) and in the conclusions (lines 431-433).

The major weaknesses of the article is that authors do not clearly explain the motivation behind their research work. Specifically, it is not clear the statement at row 22: "the services they provide are often too raw". Furthermore, the motivations scenarios reported in section 2 justify the use of Big data platforms but not the need for a new layer such a middleware on top of the Big data platforms.

In addition, the use cases under section 4 do not demonstrate the real benefits deriving from the implementation of RAM3S (except for reducing the number of lines of code as written on line 287) against the drawback to create manage the complexity of a new layer.

For the above reasons, I advise the authors to add more details about the complexity of Big Data Platforms that requires to should be managed with higher-level middleware.

R: We apologize for having been somewhat unclear in the first submitted version of the paper. We have followed the reviewer's advice by stressing out the fact that RAM^3S helps the user in implementing her data stream analysis technique without having to deal with details specific to distributed computing and Big Data platforms. In the revised version, this has been explicited in the Introduction (lines 41-42), in the Motivation (lines 108-111), in the description of the framework classes (lines 261-266), and in commenting the saving of lines of code (lines 300-306).

Finally, the authors should also specify if there other solutions available in the literature that aim to face the issue of wrapping the underlying Big Data platforms.

R: To the best of our knowledge, RAM^3S is the first middleware software to help researchers in "wrapping the underlying Big Data platforms". We have highlighted this in lines 40-42 of this revised version.

Reviewer 2 Report

The authors show how RAM3S framework can be used to implement a variety of applications independent of the particular stream processing big data platform used. The authors implement their methodology on four different applications (face recognition, plate recognition, Printed Text Recognition in Videos and Shoe  Classification ). Overall the paper is well written and organized. The idea and results are presented very clearly. I hova no additional comments on the manuscript. Hence I recommend accepting the paper.

Author Response

The authors show how RAM3S framework can be used to implement a variety of applications independent of the particular stream processing big data platform used. The authors implement their methodology on four different applications (face recognition, plate recognition, Printed Text Recognition in Videos and Shoe  Classification ). Overall the paper is well written and organized. The idea and results are presented very clearly. I hova no additional comments on the manuscript. Hence I recommend accepting the paper.

R: We thank the reviewer for appreciating our work.

Round 2

Reviewer 1 Report

The authors have addressed all the previous reviewers' comments.
Congratulations to the authors for the proposed architecture which seems very powerful.

Please consider to include the following references:

1) At pag. 1 row 17, after the following sentence: "The uses are countless, from the security of citizens [16], to sentiment analysis [13] and narrative inquiry [15]."

the authors could reference the following paper:

Modoni, G. E., & Tosi, D. (2016, August). Correlation of weather and moods of the Italy residents through an analysis of their tweets. In 2016 IEEE 4th International Conference on Future Internet of Things and Cloud Workshops (FiCloudW) (pp. 216-219). IEEE.

which introduces the use of social network data to predict moods.

2) At pag. 1 row 32, after the following sentence: "To allow efficient analysis of massive MM streams, we advocate scaling out the underlying system by using platforms for Big Data management, as opposed to scaling up the system by exploiting faster,
larger, better hardware."

the authors could reference the following paper:

Singh, D., & Reddy, C. K. (2015). A survey on platforms for big data analytics. Journal of big data, 2(1), 8.

Author Response

We included both references, as suggested by the reviewer.